# Mediterranean Milk Ladder: Integrating a Healthy Eating Plan While Reintroducing Cow’s Milk

**DOI:** 10.3390/children10020234

**Published:** 2023-01-28

**Authors:** Emilia Vassilopoulou, Colleen McMilin, Carina Venter

**Affiliations:** 1Department of Nutritional Sciences and Dietetics, International Hellenic University, 57400 Thessaloniki, Greece; 2Reckitt, Mead Johnson Nutrition Institute, Glenview, IL 60026, USA; 3Section of Allergy and Immunology, University of Colorado School of Medicine, Aurora, CO 80045, USA

**Keywords:** tolerance, food ladder, healthy eating, Mediterranean diet, milk allergy, milk ladder

## Abstract

The process of gradually reintroducing food allergens into an individual’s diet is referred to as a food allergen “ladder”, and the most recent edition of the original Milk Allergy in Primary (MAP) Care Guidelines, as well as the International Milk Allergy in Primary Care (IMAP), includes a shortened, improved, and international version with specific recipes, indicating the exact milk protein content, as well as the duration of heating and the temperature for each step of the ladder. Food allergen ladders are being used increasingly in clinical practice. The aim of this study was to develop a Mediterranean milk ladder based on the principles of the Mediterranean eating pattern. The protein content delivered in a portion of the final food product in each step of the ladder in the Mediterranean version corresponds to that provided in the IMAP ladder. Different recipes for the various steps were provided to increase acceptability and variety. Quantification of the total milk protein, casein content, and beta-lactoglobulin by Enzyme-linked immunosorbent assay (ELISA) could detect the gradual increase in concentrations, but the accuracy of the method was affected by the presence of the other ingredients in the mixtures. When developing the Mediterranean milk ladder, a key consideration was to reduce the amount of sugar by using limited amounts of brown sugar and substituting sugar with fresh fruit juice or honey for children aged older than one year. The proposed Mediterranean milk ladder includes principles of (a) healthy eating based on the Mediterranean diet and (b) the acceptability of foods across different age groups.

## 1. Introduction

The EuroPrevall birth cohort study spanning nine European countries indicated that 0.54% of children present a challenge-proven cow’s milk allergy (CMA) [1]. National incidences are reported to range from 1% (in the Netherlands and United Kingdom) to <0.3% (in Lithuania, Germany and Greece) [1]. Children diagnosed with IgE and non-IgE mediated CMA generally have a good prognosis, with successful reintroduction of particularly baked forms of milk in the first five years of life [2,3]. Delaying the reintroduction of cow’s milk can lead to a decrease in the quality of life of both the children and their families, as well as impairment of the children’s physical health [4,5,6]. 

The timing and method of reintroduction of cow’s milk is based on the type of allergy (IgE or non-IgE), clinical presentation, history of ingestion or accidental ingestion, and laboratory findings, when relevant [7,8,9]. The process of gradual reintroduction of food allergens into an individual’s diet, in order to facilitate the development of natural tolerance, is referred to as a food allergen “ladder” and was historically used in mild to moderate non-IgE mediated food allergies only [8,10], but more recent data support its safe use in IgE mediated food allergies as well [11]. 

Food allergen ladders are being used increasingly in clinical practice, when it is assumed that the allergy is resolving, but the levels of allergens in many cases are uncharacterized, and probably depend on the degree of heating and/or processing of the milk [12]. Caretakers should consult their physician and/or dietitian for reintroduction guidance, including recipes, steps of reintroduction, and identification and treatment of allergic symptoms, as well as for advice on the importance of carrying rescue medication in the case of IgE mediated food allergies [13,14,15]. In addition to the currently available ladders, there is a need for standardized food ladders that include culturally appropriate, healthy food items for successful and safe introduction of food allergens, depending on the region [16,17]. 

For reintroduction of milk, specifically, although it is generally considered safe to carry out this process outside of the office or hospital setting, and, in some countries [18,19,20], this demands a careful selection of patients based on their type of allergy, specific IgE levels, and prognosis of resolution [21]. The original Milk Allergy in Primary Care (MAP) Guidelines were transformed into the International Milk Allergy in Primary Care (IMAP) Guidelines in 2017, which is a shortened, improved, and international version of the guidelines, with specific recipes, indicating the exact milk protein content, and the recommended duration and temperature of heating for each step [8]. 

The aim of the current study was to develop a Mediterranean milk ladder based on the principles of the Mediterranean eating pattern as a healthy version of the IMAP for milk reintroduction in the Mediterranean region.

## 2. Materials and Methods

### 2.1. Recipes Development

Based on a bilingual Mediterranean recipe book dedicated to people with food allergy [22], we aimed to develop recipes with local ingredients that are representative of the Mediterranean diet, including whole grain flour, brown rice, olive oil, fresh and dried fruit, legumes, and vegetables. 

The initial design and selection of the recipes for the ladder was based on their total milk protein content, as well as the temperature and duration of heating. At least two different recipes were prepared for each step in an attempt to increase the variety and the acceptability and for users. The protein content delivered in a portion of the food products finally selected for each step of the ladder corresponded to that provided in the IMAP ladder. 

### 2.2. Milk Protein Content Analysis

Two samples from each recipe were analyzed for their total milk protein content with the Veratox Total Milk Allergen Enzyme-linked immunosorbent assay (ELISA) kit for casein with Veratox Casein Allergens ELISA Kit and for beta-lactoglobulin by SENSISpec ELISA Beta-Lactoglobulin Eurofins, provided by the Food Allergens Laboratory (www.foodallergenslab.com, accessed on 10 September 2022). The aim was to establish both the total milk protein content and the levels of the components that are denatured by heating, i.e., beta-lactoglobulin and the stable components, i.e., casein in our recipes [8,9].

## 3. Results

The recipes, as shown in Table 1, were created, based on the principal steps outlined above, to reflect culturally appropriate foods, and they contain local ingredients highlighted in the traditional Mediterranean diet. These ingredients include: vegetables, such as carrots, tomatoes and onions; fresh fruit, such as banana and dried fruit, such as raisins; whole grains, such as whole meal flour, oats and brown rice; and non-animal protein sources, such as lentils. 

Saturated fats (butter) and trans-fats (margarines) were avoided. Avocado and olive oil, key elements of the Mediterranean diet, rich in monounsaturated fatty acids, flavonoids, and polyphenols, were used in the recipes. Sugar and refined carbohydrates were used sparingly. A seven-step milk ladder was created, including the last step with the pasteurized milk. The recipes included thirteen foods for increased dietary variety during milk re-introduction, with alternatives to increase the acceptability and palatability for the users. 

Quantification of the total milk protein, casein content, and beta-lactoglobulin with ELISA was able to detect the gradual increase in the concentrations, but the accuracy of the method was affected by the presence of the other ingredients in the mixtures, such as wheat, grains, and fat (Table 2).

## 4. Discussion

We developed a Mediterranean milk ladder that includes foods based on the principles of the Mediterranean eating pattern [24,25,26]. By developing a milk ladder rooted in the traditions of the Mediterranean eating pattern, not only are we providing a format for the reintroduction of milk at home, but we are doing so in a way that includes local foods that have healthy attributes associated with prevention of chronic non-communicable diseases [24,25,26,27].

A standardized approach was used, including three key components: the food allergen, nutritional factors, and medical aspects [11]. 

The milk ladder was designed by following the principal step outlined in the IMAP ladder. We considered the dose of milk protein required and the amount of milk needed to reach the appropriate dose for each step, followed by the presence of a wheat matrix for the initial steps, as well as the duration and temperature of heating of the milk in the preparation of the recipe. The recipes also needed to be simple in preparation and contain ingredients that would be culturally accepted and familiar to the users.

The original MAP ladder, when evaluated by users and their carers, which produced criticism of the high amounts of sugars and saturated fat used in the recipes [28]. These were in contrast to healthy eating guidelines, which promote high amounts of fruit, vegetables, and whole grains, as well as monounsaturated and polyunsaturated fats rather that saturated fats. When developing the Mediterranean milk ladder, a key consideration was reduction of the amount of sugar, which was achieved by using limited amounts of brown sugar and substituting sugar with fresh fruit juice or honey for children older than one year of age [29]. As the ladder is being used in a younger population, we also wanted to ensure that the taste and texture of the foods included would be palatable to the target population while continuing to meet the key components of Mediterranean eating pattern.

The Mediterranean milk ladder is intended to be used for children with milk allergy under the supervision of a healthcare professional. The concept of milk-reintroduction through the ladder is similar to that used in the original IMAP [8]. Briefly, the healthcare professional decides when it is appropriate for the child for milk re-introduction based on his/her clinical history and symptoms. In collaboration with the parent/carer, the healthcare professional must identify the appropriate step at which the individual child should start, as some children may already include one or more forms of treated milk in their diet, for instance small portions of baked milk in cake or cookies. Before starting the ladder, the child must feel well and be free of abdominal symptoms, bowel symptoms, and eczema. 

Although the Mediterranean milk ladder recommended here consists of seven steps, the healthcare professional may adjust the number of steps or the time spent on each step. The estimated portion size in each step is provided as a guide for preschoolers [11], but this may need personalized modification based on the age and development of the individual child. Finally, when a child tolerates a food at any particular step, he/she should continue to consume this food in addition to foods from the previous steps. Conversely, when a food is not tolerated, the child should go back to the previous tolerated step, and the healthcare professional should advise on the time that this step can be tried again.

## 5. Conclusions

CMA is among the most common of food allergies worldwide. We propose here a Mediterranean milk ladder, which includes principles of (a) healthy eating based on the Mediterranean diet, as well as (b) acceptability of foods across different ages based on regional eating habits. We support the concept that this model of milk introduction provides the dietary variety needed for continuous consumption of milk during each step of the ladder, along with a healthy eating pattern that has immunomodulatory effects, which might play a role in the mitigation of the allergy outcomes [30,31]. Investigation of future application in clinical practice is needed to provide supportive evidence. 

## Figures and Tables

**Table 1 children-10-00234-t001:** Recipes for the Mediterranean milk ladder.

**1.1** Beef burger.	
**Ingredients**	
500 g	minced beef (15% fat) or chicken burger
22 g (0.7 g protein)	milk full fat (3.2 g protein/100 mL)
100 g	oat flakes
78 mL	olive oil
1 tbsp	oregano
100 g	onion finely chopped
	salt and pepper
800 g	Total weight
**80 g raw/~60 g baked**	**Served portion**
**Preparation**	
1. Combine all the ingredients in a large bowl. 2. Knead and place the mixture in the fridge for 30 min. 3. Shape in six burgers of 80 g each (raw material). 4. Preheat oven at 200 °C. 5. Bake in a preheated at 200 °C for 30 min.	
**1.2** Oat cookies with olive oil.	
Ingredients	
300 g	oats flour (oats ground in blender)
90 g	honey
40 g	raisins
70 g	olive oil
1 tsp	baking soda
1 pinch	salt
2 tbsp	chopped flaxseed (optional)
22 mL (0.7 g milk protein)	milk 2% fat (3.2 g protein/100 mL)
a pinch of	cinnamon
**550 g**	**Total weight**
**55 g raw/40 g baked cookie**	**Served portion**
**Preparation**	
1. Grind the oats in the food processor for two to three seconds until you obtain a fine flour. 2. Add all the other ingredients, except the milk and mix until smooth, for about 10 s. 3. Add the milk, slowly until the dough forms. 4. Take the dough out of the processor. Shape with your hands into a ball. Wrap it in cling film and refrigerate for about 1 h. 5. Preheat the oven to 180 °C, in the air setting 6. Cover a baking tray with non-stick paper. 7. Form 10 small balls with your hands and press them in the middle. 8. You will make 10 cookies of 55 g each (~40 g raw). 9. Bake for about 12–15 min, until they turn light brown at the edges. 10. Take them out of the oven and let them cool down completely.	
**2.1** Sweet whole wheat muffins with berries or raisins.	
**Ingredients**	
250 g	whole grain flour
1 tsp	baking soda
75 g	brown sugar
100 g	berries or raisins
125 mL (4 g milk protein)	full fat milk (3.2 g/100 mL)
10 mL	olive oil
25 mL	water
**585 g**	**Total weight**
**125 g raw (0.875 g milk protein)/100 g baked**	**Served portion**
**Preparation**	
1. Sift the flour and baking powder into a bowl. 2. Add the sugar and the raisins or blueberries. 3. Mix and make a hole in the middle. 4. Pour the milk, the olive oil and the water in another bowl and mix with a whisk. 5. Combine the liquid with the solid ingredients and mix until you obtain a homogenous mixture. 6. Fill up 6 muffin cases (2/3 full). 7. Preheat the oven at 170 °C. 8. Bake for approximately 25 min.	
**2.2** Whole wheat savory muffins.	
**Ingredients**	
250 g	whole wheat flour
1 tsp	baking soda
a pinch of	oregano
100 g	olives without the stone, in small pieces
75 g	tomato trimmed
125 mL (4 g milk protein)	full fat milk (3.2 g/100 mL)
10 mL	olive oil
25 mL	water
**585 g**	**Total weight**
**125 g raw (0.875 g milk protein)/100 g baked**	**Served portion**
**Preparation**	
1. Sift the flour and baking powder into a bowl. 2. Add the trimmed tomato, the olives, the oregano. 3. Mix and make a hole in the middle. 4. Pour the milk, the olive oil and the water in a bowl and mix with a whisk. 5. Combine the liquid with the solid ingredients and mix until you obtain a homogenous mixture. 6. Fill up 6 muffin cases (2/3 full). 7. Preheat the oven at 170 °C. 8. Bake for approximately 25 min.	
**3.1** Mediterranean type pureed potato.	
**Ingredients**	
600 g	potatoes (900 g without skin)
200 g	carrots
100 g	onion
100 g	olive oil
200 g (6.4 g milk protein)	full fat milk (3.2 g/100 mL)
	salt and pepper
	grated nutmeg
**1200 g**	**Total weight**
200 g	**Served portion**
**Preparation**	
1. Peel the potatoes and cut them into cubes, put them in a medium saucepan, and add cold water to cover. Add a little salt to the water and cover the pan. 2. Bring the potatoes to boil. Cover the pot and cook for about 35 min or until the potatoes are soft. 3. Drain the potatoes well and break them into a puree. Mix in the rest of the ingredients, olive oil, milk, salt, pepper, and nutmeg. Stir until the puree is homogenous.	
**3.2** Crepes with whole wheat flour.	
**Ingredients**	
300 g	whole wheat flour
300 g (9.6 g/milk protein)	full fat milk (3.2 g milk protein/100 mL)
15 g	olive oil
A pinch of	salt
A pinch of	brown sugar
	olive oil for frying
**620 g**	**Total weight**
95 g raw/85 g baked	**Served portion**
**Preparation**	
1. Put the flour, salt and sugar in a bowl and mix them with a whisk. 2. Add the oil and milk and mix into a batter with a whisk. 3. Allow the batter to stand for 10 min. 4. Heat a 24 or 26 cm non-stick pan 5. Add a few drops of olive oil to the pan and wipe with kitchen paper. 6. Add about 1/2 cup of batter. 7. Shake the pan by the handle so that the batter is spread all over the surface of the pan and it becomes a thin crepe. 8. Bake until the crepe comes off and hardens. About 2–1/2 min 9. Flip the crepe over and cook on the other side for about 1–1/2 min. 10. Remove the crepe to a plate and continue making the rest of the crepes. 11. Serve with topping of preference, such as fresh fruit, homemade jam, honey, or other ingredients.	
**4.1** Bread with cheese, olives, and tomato.	
**Ingredients**	
480 g	wholewheat flour
400 g (56 g milk protein)	feta cheese
150 g	tomato
50 g	olives without the stone
200 g	lukewarm water
70 mL	olive oil
1 package (7 g)	dry yeast
1/4 tsp	salt
A pinch of	oregano
**1350 g**	**Total weight**
~80 g raw portion/55 g baked	**Served portion**
**Preparation**	
1. In a bowl, knead all the ingredients together well or mix in a mixer until you have a homogenous, non-sticky, dough. 2. Preheat the oven at 36 °C 3. Create a loaf of bread and transfer the dough onto a baking tray. 4. Cover with a cloth and put in the oven for about an hour to allow to the loaf to rise. 5. Preheat the oven at 180 °C. 6. Bake for ~one hour. Do not open the oven during the baking.	
**4.2** Lentil burger with cheese.	
**Ingredients**	
175 g	lentils (dry, not boiled)
1 piece	laurel leaf
100 g	oats
60 g	wholewheat flour
100 g	onion
1 clover	garlic
100 g (25 g milk protein)	grated hard cheese (e.g., regato, kefalotyri)
1/4 tsp	paprika powder
90 g	olive oil
1 tbsp	parsley
	Salt, pepper
625 g	**Total weight**
85 g raw/60 g baked	**Served portion**
**Preparation**	
1. Boil lentils until soft for about 40 min in medium heat together with the laurel leaf. 2. Drain the lentils and dry them in a cloth. 3. Pour them into a large bowl. 4. Place the onion, garlic, cumin, pepper, paprika, and parsley in the food processor and mix until smooth. 5. Mix the onion mixture with the lentils. 6. Add the grated cheese and mix. 7. Cover the bowl and chill in the fridge for an hour. 8. Shape into small balls and place them in a baking tray. 9. Bake in a preheated oven at 180 °C for 50 min.	
**5.1** Rice pudding.	
**Ingredients**	
100 g	brown rice
300 g	water
50 g	honey or brown sugar
600 mL (19.2 g milk protein)	full fat milk (3.2 g milk protein)
1/2 tbsp	ground mastic or vanilla
20 g	corn flour
1/8 tsp	salt
A pinch of	cinnamon
**1070 g**	**Total weight**
960 g	weight after cooking
170 g	**Served portion**
**Preparation**	
1. Wash the rice and drain it. 2. Put the water in a saucepan to boil and add the salt. 3. When it boils, add the rice. 4. Simmer for 15 min until the rice pops. 5. Then, add the milk, sugar, and mastic or vanilla and mix. 6. When the rice is soft, dissolve the corn flour in half a cup of cold water and while stirring, pour it into the pot. 7. Simmer for a while until it starts to coagulate. 8. Divide the mixture into six serving bowls of 170 g each. 9. Serve warm or cold. 10. Leave it to cool well before storing it in the fridge. 11. Sprinkle with cinnamon before serving if you wish.	
**5.2** Bechamel.	
**Ingredients**	
1000 mL (32 g milk protein)	full fat milk (3.2 g/100 mL)
120 g	olive oil
120 g	rice flour or corn flour or whole wheat flour
	salt, pepper, and nutmeg powder
**1240 g**	**Total weight**
1200 g	weight after cooking
130 g	**Served portion**
**Preparation**	
1. Mix olive oil and flour in a pot. 2. Add milk and stir constantly for 20 min until it starts to boil. 3. Turn off the fire. 4. Add rice or corn flour and stir until it becomes thick and continue to stir until it becomes thick.5. Add salt, pepper, and nutmeg powder for flavor. 6. Serve on top of vegetables, pasta, or rice.	
**6**. Yogurt.	
**Ingredients**	
2000 mL	cow’s milk (64 g protein)
200 g (1 cup) (6.7 g milk protein)	full fat traditional sheep yogurt (12 g)
**190 g**	Served portion
**Preparation**	
1. Heat the milk on medium heat, stirring continuously so that it does not stick, at 90 °C if it is not pasteurized, or at 52 °C if it is pasteurized. You will need a kitchen thermometer, but if you do not have one, you can turn off the heat just before the unpasteurized milk starts to boil, that is, as soon as we see it that it starts to swell or for the pasteurized milk when it starts to steam (~52 °C). 2. Dissolve the traditional sheep yogurt without the skin in a bowl in 10 tablespoons of warn milk from the pot, so that it becomes smooth, without pieces, then pour it into the rest of the milk and stir the milk slightly. 3. Place the pot immediately after in a warm place, cover it with a clean towel and then a lid, and leave it still for three to five hours. Alternatively, you can put it uncovered in the oven, preheated to 45–50 °C. Caution: do not shake or move the utensil during coagulation. Leave it for at least two to three hours without moving it. When it is ready, place the yogurt in the refrigerator, and consume it within 10 days.	
**7.1** Cocoa-banana ice-cream.	
**Ingredients**	
250 g	ripe bananas
50 g	cocoa powder
400 mL (12.8 g milk protein)	full fat milk (3.2 g protein/100 mL)
700 g	Total weight
190 g	Served portion
**Preparation**	
1. Slice the bananas and put them in a bowl. 2. Place them in the freezer and allow to freeze. 3. Put the bananas in a blender. Add the cocoa and the milk. 4. Process until well blended. 5. Empty the mixture into a bowl and with an electric mixer beat for a few minutes until smooth. 6. Freeze for two hours. Take out and beat with the mixer again. 7. Repeat the same procedure two or three times (freezing for two hours and then beating) until the ice cream is smooth and fluffy without crystals. Alternatively, you can use strawberries, cherries, melon, watermelon, apricots, or peaches. If you wish, honey may be added.	
**7.2** Avocado-cocoa mousse.	
**Ingredients**	
220 g	very ripe and soft avocadoes cut into pieces
40 g	cocoa powder
100 g	honey
140 g (9.8 g milk protein)	evaporated milk (7 g protein/100 g)
1 tsp	vanilla extract
8 drops	fresh lemon
	topping milk-free chocolate grated
500 g	Total weight
175 g	portion
**Preparation**	
1. Select very ripe, very soft avocados, so that the mousse has a velvety texture. If it is harder, the avocado will not melt, and the mousse will not become smooth and uniform. 2. Add the lemon drops to prevent oxidation and discoloration. 3. Put all the ingredients in the blender and beat until the mixture is uniform and the chocolate mousse with avocado is formed. 4. Divide into tall glasses. 5. Spread the grated milk free chocolate on top. 6. Serve immediately or keep refrigerated.	

**Table 2 children-10-00234-t002:** Amounts of milk and cooking temperatures in the Mediterranean milk ladder recipes in comparison to the IMAP milk ladder.

		Energy Value (Kcal)/100 g	Portion (g) (*)	Estimated Milk Protein (g/Portion)	IMAP (g/Portion)	Temperature (°C)	Cooking Time (Minutes)	Milk Protein (ELISA Sandwich) g/Portion	Casein (ELISA) mg/Portion (**)	Beta-Lactoglobulin (ELISA) mg/Portion (**)
1.1	Beef burger	108	~60 g baked (80 g raw)	0.07	0.07	180	30	0.12	0.01	0.05
1.2	Oat biscuit with olive oil	95	~40 g baked (55 g raw)	0.07	0.07	180	12 to 15	0.20	0.09	0.05
2.1	Sweet whole wheat muffins with berries or raisins	137	100 g baked (125 g raw)	0.875	0.875	170	25	0.48	0.31	0.41
2.2	Savory muffin with tomato and olives	135	100 g baked (125 g raw)	0.875	0.875	170	25	0.52	0.38	0.48
3.1	Med type potato puree	145	200 g cooked	1.47	1.47	~50 (add cold milk to hot mixture of 100 °C)	n/a	1.82	0.68	0.20
3.2	Crepes with whole wheat flour	180	85 g baked	1.47	1.47	100	5	1.76	0.97	0.75
4.1	Bread with cheese, olives and tomato	160	55 g baked (82 g raw)	3.44	3.44	180	60	2.64	1.6	1.8
4.2	Lentil burger with cheese	110	60 g baked (85 g raw)	3.44	3.44	180	30	2.44	1.80	0.38
4.3	Feta cheese 14 gr milk protein/100gr	264	24.5 g	3.44	3.44	65	24 h–3 months to consume	n/a	n/a	n/a
5.1	Rice pudding	134	170 g cooked	3.44	3.44	100	20	16	61	7.8
5.2	Bechamel	127	130 g cooked	3.44	3.44	100	20	13	2.38	6.16
6	Yogurt	140	100 g	6.7	6.7	52 or 90	30	6.8	22.21	5.01
7.1	Cocoa banana ice	75	190 g	3.44	3.44	n/a	n/a	2.45	1.46	0.95
7.2	Avocado-cocoa ice	162	175 g	3.44	3.44	n/a	n/a	3.58	1.12	0.89
7.3	milk	73	100 g	3.2	3.2	n/a	n/a	n/a	n/a	n/a

* estimated portion sizes for preschoolers [23]; ** reduction in casein and beta-lactoglobulin levels may not indicate that the allergens are destroyed but may indicate reduced bioaccessibility. n/a: non-applicable. *** ELISA: Enzyme-linked immunosorbent assay.

## Data Availability

Not applicable.

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
