# Peer review of "Mediterranean Milk Ladder: Integrating a Healthy Eating Plan While Reintroducing Cow’s Milk"

_children, 2023, doi:10.3390/children10020234_

Round 1
Reviewer 1 Report
This communication proposed the Mediterranean milk ladder based on the Mediterranean diet and people with cow’s milk allergy at different ages. However, there is confusion about how to apply these recipes to the corresponding population’s diet. In addition, many formatting and writing errors need to be modified.
Specific Comments:
1. There are many mistakes in the reference format. The format of references in the text should be uniform and in accordance with the requirements of the journal of Children. As follows:
1. Author 1, A.B.; Author 2, C.D. Title of the article. Abbreviated Journal Name Year, Volume, page range.
2. The unit of “ml” should be “mL” in this article.
3. Line 38 and line 41
“Ige” and “igE” should be modified to “IgE”
4. Line 104 and line 74
“b-lactoglobulin” should be modified to “beta-lactoglobulin”
5. Line 71 and table 2
“Elisa” should be changed to “ELISA”
6. In table 6, “45 °- 50 ℃” should be “45- 50 ℃”
7. Line 110-111
The annotation of n/a in Table 2 needs to be added to the table.
8. There is an extra space between the number and unit, such as “0.7gr protein” should be “0.7 gr protein” in table 1.1. Please check the whole article carefully.
Author Response
Reviewer 1
This communication proposed the Mediterranean milk ladder based on the Mediterranean diet and people with cow’s milk allergy at different ages.
We appreciate your overall comment.
However, there is confusion about how to apply these recipes to the corresponding population’s diet.
We have incorporated a relevant text in our document to facilitate understanding (Lines 183-199). We hope it is according to your expectations.
In addition, many formatting and writing errors need to be modified.
A thorough text editing has been performed to avoid such errors. Thank you for your helpful comment.
Specific Comments:
- 1. There are many mistakes in the reference format. The format of references in the text should be uniform and in accordance with the requirements of the journal of Children. As follows:
- Author 1, A.B.; Author 2, C.D. Title of the article. Abbreviated Journal Name Year, Volume, page range.
The reference list is checked and updated accordingly
- The unit of “ml” should be “mL” in this article.
Units are corrected
- Line 38 and line 41
“Ige” and “igE” should be modified to “IgE”
Mistyping is corrected
- Line 104 and line 74
“b-lactoglobulin” should be modified to “beta-lactoglobulin”
Mistyping is corrected
- Line 71 and table 2
“Elisa” should be changed to “ELISA”
Mistyping is corrected
- In table 6, “45 °- 50 ℃” should be “45- 50 ℃”
Corrected
- Line 110-111
The annotation of n/a in Table 2 needs to be added to the table.
Added
- There is an extra space between the number and unit, such as “0.7gr protein” should be “0.7 gr protein” in table 1.1. Please check the whole article carefully.
Checked and corrected. Thank you
Reviewer 2 Report
CMA is among the most common food allergies worldwide. From 2017, doctors and nutritionists can use the guidelines on gradually reintroducing food allergens (particularly baked forms of milk) into an individual's diet of patients with cow's milk allergy and other food allergens. The guidelines mainly apply to patients with mild to moderate non-IgE mediated CMA and other food allergies. Recently is suggested to use this guidelines in IgE-mediated food allergy too. Recommendations are prepared in the form of food allergen “ladders”: Milk Allergy in Primary Care Guidelines (MAP 2017), and the latest edition - International Milk Allergy in Primary Care (iMAP 2019). International version contains specific recipes, indicating the exact milk protein content, time of heating and temperature of heating for each step.
The Italian authors adapted these guidelines to develop a Mediterranean milk ladder based on the principles of the Mediterranean eating pattern. The authors have developed a seven step milk-ladder included thirteen foods for providing more options during milk reintroduction to increase acceptability and palatability to the users. The Mediterranean milk ladder meets the conditions of: principles of healthy eating based on the Mediterranean diet, acceptability of foods proposed across different ages.
The publication is very valuable and useful in clinical management of patients with CMA, and other food allergies.
Author Response
We feel honored by your positive comment. We appreciate it as we really feel that by applying an overall healthier diet, even while reintroducing milk through the milk ladder we can gradually develop a healthier Mediterranean (Greek actually, but Italy is also great) “substrate” for halting immune-related disease such as food allergies in children. Thank you very much.
Reviewer 3 Report
Dear Editor and Authors,
I send you my review about the short communication “Mediterranean Milk Ladder: Integrating a healthy eating plan while reintroducing cow’s milk”.
The scope of the paper, as reported in the aim was to develop a Mediter-ranean milk ladder based on the principles of the Mediterranean eating pattern.
In my opinion, the communication do not result well structured and although it is well written, it need of a little revision of the English language. In particular the use of some phrasal verbs, like for examples “build-up”, should be avoid.
Furthermore, in the present form, this communication show also others lacks, that I report below.
The introduction do not adequately support the aim of the study and it result to much general. I suggest to the Authors to made a little comparison with the previous articles reported in literature and to stress the difference among them and their current study.
The paragraph materials and methods needs to be improved adding the number of the trials made and the number of replicates of the analysis.
Furthermore, since the composition of the recipes are decided by the researchers the tables from 1.1 to 7.2 should be shift from the chapter “Results” to the chapter “Materials and Methods”.
The results is not completely presented, they are not deep discussed and they are not compared to the data reported in the literature.
Moreover, line 126 the Authors report that “Based on the feedback received by the consumer…” but in the chapter of methods they do not described how this feedback are collected and how this data were analysed. To improve the discussion I suggest to the Authors to deepind this topic.
Finally, the conclusions should be improved and the authors should suggest the benefit of their ladder.
Best regards
Author Response
Dear Editor and Authors,
I send you my review about the short communication “Mediterranean Milk Ladder: Integrating a healthy eating plan while reintroducing cow’s milk”.
The scope of the paper, as reported in the aim was to develop a Mediterranean milk ladder based on the principles of the Mediterranean eating pattern.
Thank you for reviewing our work.
In my opinion, the communication do not result well structured and although it is well written, it need of a little revision of the English language. In particular the use of some phrasal verbs, like for examples “build-up”, should be avoid.
A native English speaker has revised our manuscript to avoid such errors.
Furthermore, in the present form, this communication show also others lacks, that I report below.
The introduction does not adequately support the aim of the study and it result to much general. I suggest to the Authors to make a little comparison with the previous articles reported in literature and to stress the difference among them and their current study.
The paragraph materials and methods need to be improved adding the number of the trials made and the number of replicates of the analysis.
The introduction and the material and section methods have been enriched and we hope changes are according to your expectations.
Furthermore, since the composition of the recipes are decided by the researchers the tables from 1.1 to 7.2 should be shift from the chapter “Results” to the chapter “Materials and Methods”.
Following the changes on the methods section, we believe that we have enhanced our document appropriately, so as to avoid moving the recipes in the methods section. To our opinion it is important to have the recipes in the results section, as this is the deliverable of the Mediterranean milk ladder, for further clinical practical use.
The results is not completely presented, they are not deep discussed and they are not compared to the data reported in the literature.
Moreover, line 126 the Authors report that “Based on the feedback received by the consumer…” but in the chapter of methods they do not describe how this feedback are collected and how this data were analysed. To improve the discussion, I suggest to the Authors to deep in this topic.
In this line we refer to the feedback for MAP ladder and not the Mediterranean milk ladder.
Additional efforts to deep in the topic are applied- hopefully providing an optimum result.
Finally, the conclusions should be improved, and the authors should suggest the benefit of their ladder.
The conclusion section is modified.
Thank you for your remarks
Reviewer 4 Report
Dear authors, the basic theme of the article is an interesting one, but the method of making the article unfortunately has a weak scientific value. It looks more like a gastronomic guide than a scientific work. I believe that a total overhaul is necessary in which the caloric value of the food plans should also be highlighted, and interestingly, it should also be correlated with the beneficial effects observed during implementation, i.e. a complete study. The discussions are inappropriate, as are the conclusions and bibliographic documentation. It is an article suitable for an information magazine, not quoted web of science as it is presented in its current form. I propose the complete restoration.
Author Response
We appreciate your overall comment. We understand your second thoughts about its content, but we really believe that science starts from changes originating and applied in real life. Milk allergy is among the most common food allergies in children and timely reintroduction with appropriate food matrices facilitates both the resolve of symptoms and the quality of life of patients.
By modifying the way that we prepare foods we also modify our immune system’s response through changes in inflammatory indices and microbiome. Although the final outcomes are important after applying these protocols are more of a scientific value, the “know how to do it” with appropriate simple procedures in everyday life and clinical practice are lacking scientific literature.
We appreciate your points, and under this perspective, the current work is submitted as a communication and not as a full article. Of course, further application is indented in the clinical practice, but we really believe that such practical guidance is useful for the scientific community to proceed in wider findings in clinical practice.
We do hope of your reconsideration of our manuscript, also by taking into account that we have made a thorough revision of our manuscript based on the valuable comments of the other reviewers.
Kindest, the authors.
Round 2
Reviewer 1 Report
It can be accepted now.
Author Response
Thank you very much!
Reviewer 3 Report
In this version the article result well written and sufficiently well structured.
Overall, the new version of the short communication resulted improved
Well done
Author Response
Thank you so much!
Reviewer 4 Report
The article has been improved, but no food plan can be complete without a correct calculation of the energy value per 100 g prepared.
Author Response
The calculation of the energy value per 100gr is added on Table 2.
Thank you for your efforts to improve our work.
Round 3
Reviewer 4 Report
If the editor considers it a suitable article for the journal, then it can be published. The article has been improved, but I think it could be even better.